

# Effects of step length and cadence on hip moment impulse in the frontal plane during the stance phase

Takuma Inai[1], Tomoya Takabayashi[2], Mutsuaki Edama[2] and Masayoshi Kubo[2]

[1] Exercise Motivation and Physical Function Augmentation Research Team, Human Augmentation Research Center, National Institute of Advanced Industrial Science and Technology, Kashiwa City, Japan
[2] Institute for Human Movement and Medical Sciences, Niigata University of Health and Welfare, Niigata City, Japan

## ABSTRACT

**Background**. An excessive daily cumulative hip moment in the frontal plane (determined as the product of hip moment impulse in the frontal plane during the stance phase and mean number of steps per day) is a risk factor for the progression of hip osteoarthritis. Moreover, walking speed and step length decrease, whereas cadence increases in patients with hip osteoarthritis. However, the effects of step length and cadence on hip moment impulse in the frontal plane during the stance phase are not known. Therefore, this study aimed to examine the effects of step length and cadence on hip moment impulse in the frontal plane during the stance phase.

**Methods**. We used a public dataset (kinetic and kinematic data) of over-ground walking and selected 31 participants randomly from the full dataset of 57 participants. The selected participants walked at a self-selected speed and repeated the exercise 15 times. We analyzed the data for all 15 trials for each participant. Multiple regression analysis was performed with the hip moment impulse in the frontal plane during the stance phase as the dependent variable and step length and cadence as independent variables.

**Results**. The adjusted $R^2$ in this model was 0.71 ($p < 0.001$). The standardized partial regression coefficients of step length and cadence were 0.63 ($t = 5.24$; $p < 0.001$) and $-0.60$ ($t = -4.58$; $p < 0.001$), respectively.

**Conclusions**. Our results suggest that low cadence, not short step length, increases the hip moment impulse in the frontal plane. Our findings help understand the gait pattern with low hip moment impulse in the frontal plane.

Corresponding author
Takuma Inai, hwd17001@nuhw.ac.jp

## INTRODUCTION

Degeneration of the hip articular cartilage is noted in patients with hip osteoarthritis, which reduces their physical function ability. Patients with hip osteoarthritis experience hip joint pain (*Iidaka et al., 2016*; *Iidaka et al., 2020*), and decrease in muscle strength (*Loureiro, Mills & Barrett, 2013*; *Zacharias et al., 2016*; *Loureiro et al., 2018*) and range of hip joint motion (*Holla et al., 2011*; *Holla et al., 2012*). Such changes decrease the patients' ability to perform activities of daily living (*Pisters et al., 2012*; *Pisters et al., 2014*) and reduces their

quality of life (*Salaffi et al., 2005*; *Boutron et al., 2008*). It is, therefore, important to prevent the progression of hip osteoarthritis to avoid these problems.

*Tateuchi et al. (2017)* proposed a new index in the frontal plane called daily cumulative hip moment (calculated as the product of hip moment impulse in the frontal plane and mean number of steps per day) and examined the relationship between this index and the progression of hip osteoarthritis (i.e., width of the hip joint space). They found that an excessive daily cumulative hip moment in the frontal plane is a risk factor for narrowing of the hip joint space width (*Tateuchi et al., 2017*). Therefore, it is important to avoid excessive daily cumulative hip moment in the frontal plane to prevent narrowing of the hip joint space width.

Considering that the walking speed is lower in individuals with hip osteoarthritis than in those without hip osteoarthritis (*Constantinou et al., 2017*; *Foucher, 2017*; *Meyer et al., 2018*; *Wesseling et al., 2018*; *Diamond et al., 2018*), *Inai et al. (2019a)* examined the relationship between walking speed and hip moment impulse in the frontal plane and found that a decrease in the walking speed increases the hip moment impulse in the frontal plane during the stance phase. Therefore, to avoid a high daily cumulative hip moment in the frontal plane, patients with hip osteoarthritis should maintain a normal walking speed (i.e., avoid reducing the walking speed).

Walking speed is calculated as the product of step length and cadence. *Constantinou et al. (2017)* found that step length decreases for patients with hip osteoarthritis compared to controls. *Schmidt et al. (2017)* showed that cadence is greater in patients with hip osteoarthritis increases than in controls. However, the effects of step length and cadence on hip moment impulse in the frontal plane during the stance phase are not known.

This study aimed to examine the effects of step length and cadence on hip moment impulse in the frontal plane during the stance phase. We hypothesized that (1) a decrease in step length decreases the hip moment impulse in the frontal plane and (2) a decrease in cadence increases the hip moment impulse in the frontal plane. To the best of our knowledge, this is the first study to evaluate the effects of step length and cadence on hip moment impulse in the frontal plane.

## MATERIALS & METHODS

### Participants

Data were collected as previously described in *Horst et al. (2019a)* and *Horst et al. (2019b)*. Specifically, 57 healthy adults (29 women, 28 men; age: 23.1 (2.7) years; height: 1.74 (0.10) m; body mass: 67.9 (11.3) kg) participated. Prior to the experiment, each participant read and signed a consent form that had previously been approved by the ethical committee of the Rhineland-Palatinate Medical Association in Mainz, Germany (*Horst et al., 2019a*; *Horst et al., 2019b*).

### Experimental protocol and data acquisition

The details of the experimental protocol and data acquisition were described previously (*Horst et al., 2019a*; *Horst et al., 2019b*). All participants performed 20 walking trials (self-selected speed; barefoot) on a 10-m path. Because some of the trials had technical errors, we

used 15 trials for each participant. Ten Oqus 310 infrared cameras (Qualisys AB, Sweden) captured the three-dimensional marker trajectories at a sampling frequency of 250 Hz. Three-dimensional ground reaction forces were recorded by two Kistler force plates (Type 9287CA - Kistler, Switzerland) at a frequency of 1,000 Hz. A full-body marker set consisting of 62 reflective markers placed on anatomical landmarks was used. For the static trial, all 62 reflective markers were used. For the dynamic trial (i.e., walking trial), only 54 reflective markers were used. Two experienced assessors attached the markers and conducted the analysis.

## Data processing

First, we conducted a statistical power analysis for linear regression analysis using G*Power 3.1 software (effect size $f^2 = 0.35$ [large] based on *Cohen (1992)*, significance level = 0.05, power = 0.8, number of predictors = 2), which indicated that the required sample size was 31 subjects. Therefore, we selected 31 participants randomly (15 women, 16 men; age: 23.0 [2.6] years; height: 1.76 [0.10] m; body mass: 68.8 [10.9] kg) from the 57 participants. For each participant, the data for 15 trials were randomly selected for analysis. We evaluated the stance phase of the right limb (i.e., from right heel contact to right toe-off) for each participant (*Tateuchi et al., 2017*). The three-dimensional marker trajectories, ground reaction forces, center of pressures, and moments of the force plates were filtered using a fourth-order Butterworth low-pass filter at a cut-off frequency of 6 Hz (*Inai et al., 2019b*).

The external hip adduction moment during the stance phase of the right limb was calculated for each participant using inverse dynamics (Newton-Euler method). Subsequently, the hip joint moment impulse in the frontal plane was calculated by integration of the external hip adduction moment. The hip moment impulse in the frontal plane (Nm s) was normalized by body mass (Nm s/kg). To confirm the quantitative validity of the external hip adduction moment during the stance phase, we also calculated the first peak external hip adduction moment during the first half (0–50%) of the stance phase. The first peak external hip adduction moment (Nm) was normalized with body mass (Nm/kg).

Mass, mass position, and inertia parameters of the segments reported previously (*De Leva, 1996*; *Robertson et al., 2013*) were used. The hip joint center was derived as a point interpolated at a distance of 18% of the vector norm from each reflective marker of the superior aspect of the greater trochanter along the vector (*Kito et al., 2010*). The right knee joint center was defined as the midpoint of the medial and lateral epicondyles of the right femur. The right ankle joint center was defined as the midpoint of the right medial and lateral malleoli.

The step length from the point of right heel contact was defined as the distance between the right and left heels. The step length was normalized with height (m/HT). Cadence (steps/min) was calculated using the time from right heel contact to left heel contact. Walking speed was calculated using the reflective markers placed on the sacrum and normalized with height (m/HT). Stance time was defined as the time from the time of right heel contact to that of right toe-off. All gait analyses were performed using Scilab (Scilab Enterprises, France).
**Table 1  The results of the multiple regression analysis.**

| Independent variable | Standardized partial regression coefficient | t value | p value |
|---|---|---|---|
| Step length, m/HT | 0.63 | 5.24 | <0.001 |
| Cadence, steps/min | −0.60 | −4.58 | <0.001 |

**Notes.**

Adjusted $R^2 = 0.71$. F $(2, 28) = 37.0$ $(p < 0.001)$.
Dependent variable, hip moment impulse in the frontal plane during the stance phase (Nms/kg); HT, Height.

## Statistical analysis

Multiple regression analysis was performed in which the dependent variable was defined as hip moment impulse in the frontal plane during the stance phase, and the independent variables were defined as step length and cadence. To confirm the multicollinearity of this model, the variance inflation factor was calculated. We also confirmed whether the residuals of the model followed a normal distribution, and the correlation coefficient between step length and cadence was also confirmed.

The Shapiro-Wilk test was used to determine whether the variables (step length, cadence, walking speed, stance time, and hip moment impulse in the frontal plane) followed a normal distribution. The Pearson's correlation or Spearman's correlation was used, depending upon the results of the normality test. The significance level was set at <0.05.

## RESULTS

Table 1 shows the results of the multiple regression analysis. A positive relationship between step length and hip moment impulse in the frontal plane was observed (i.e., a decrease in step length decreased the hip moment impulse in the frontal plane). An inverse relationship between cadence and hip moment impulse in the frontal plane was also observed (i.e., a decrease in cadence increased the hip moment impulse in the frontal plane). The variance inflation factor in the multiple regression analysis was 1.136, and the residuals followed a normal distribution ($p = 0.905$).

Figure 1 shows the relationship among step length, cadence, and hip moment impulse in the frontal plane. Long step length and low cadence resulted in an increased hip moment impulse in the frontal plane. In contrast, short step length and high cadence resulted in a decreased hip moment impulse in the frontal plane.

Figure 2 shows the waveform of the average external hip adduction moment during the stance phase (Nm/kg). The external hip adduction moment increased from 0% (i.e., right heel contact) to approximately 20% of the stance phase, but it decreased from approximately 80% to 100% (i.e., right toe-off) of the stance phase. Bimodal peaks were also observed in this analysis.

Table 2 presents the results for the analysis of the gait parameters. The mean and standard deviation (SD), minimum, and maximum values for each variable (hip moment impulse in the frontal plane, first peak external hip adduction moment, step length, cadence, walking speed, and stance time) are shown.

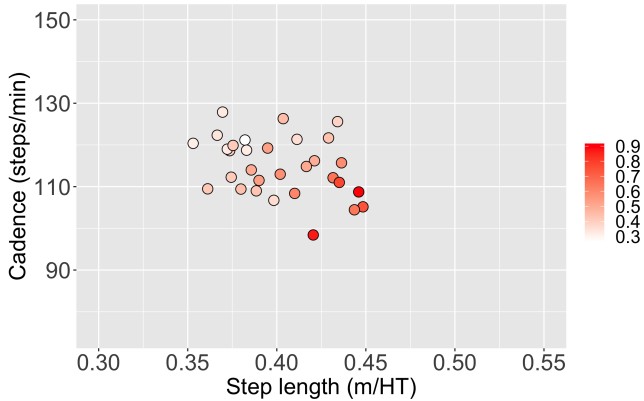

**Figure 1** **The relationship between step length, cadence, and hip moment impulse in the frontal plane during the stance phase.** Red points mean an increased hip moment impulse during the stance phase, and the shade of color indicates the quantitative values of the hip moment impulse in the frontal plane.

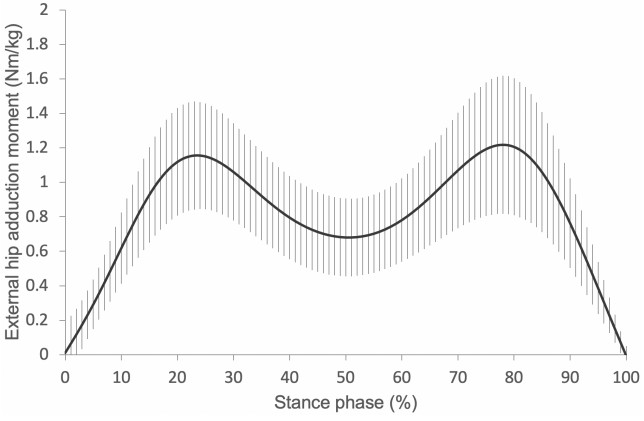

**Figure 2** **The average wave form of the external hip adduction moment during the stance phase.** A bi-modal peak was observed. Error bars mean the standard deviations.

**Table 2** **Results of the gait parameters.**

| | Mean | SD | Minimum | Maximum |
|---|---|---|---|---|
| Hip moment impulse in the frontal plane, Nms/kg | 0.50 | 0.17 | 0.27 | 0.92 |
| First peak external hip adduction moment, Nm/kg | 1.17 | 0.32 | 0.67 | 1.84 |
| Step length, m/HT | 0.40 | 0.03 | 0.35 | 0.45 |
| Cadence, steps/min | 114.9 | 7.0 | 98.4 | 127.9 |
| Walking speed, m/(sHT) | 0.82 | 0.06 | 0.73 | 0.96 |
| Stance time, s | 0.61 | 0.05 | 0.53 | 0.73 |

**Notes.**
HT, Height

**Table 3** The correlation coefficients between step length, cadence, walking speed, and stance time, and hip moment impulse in the frontal pane.

| | Cadence | Walking speed | Stance time | Hip moment impulse in the frontal plane, Nms/kg |
|---|---|---|---|---|
| Step length, m/HT | $r = -0.35, p = 0.056$ | **$r = 0.76, p < 0.001$** | $r = 0.27, p = 0.132$ | **$r = 0.73, p < 0.001$** |
| Cadence, steps/min | – | $r = 0.30, p = 0.100$ | **$r = -0.97, p < 0.001$** | **$r = -0.65, p < 0.001$** |
| Walking speed, m/(sHT) | – | – | **$r = -0.37, p = 0.040$** | $r = 0.35, p = 0.058$ |
| Stance time, s | – | – | – | **$r = 0.57, p < 0.001$** |

**Notes.**
HT, Height. Bold letters indicate a significant difference.

Table 3 presents the correlation coefficients between variables (step length, cadence, walking speed, stance time, and hip moment impulse in the frontal plane).

## DISCUSSION

This study examined the effects of step length and cadence on hip moment impulse in the frontal plane during the stance phase. Our main findings are as follows: (1) a decrease in step length decreased the hip moment impulse in the frontal plane, and (2) a decrease in cadence increased the hip moment impulse in the frontal plane. Therefore, our hypotheses were confirmed.

Patients with hip osteoarthritis exhibit decreased walking speed (*Constantinou et al., 2017*; *Foucher, 2017*; *Meyer et al., 2018*; *Wesseling et al., 2018*; *Diamond et al., 2018*) and step length (*Constantinou et al., 2017*). However, walking speed depends on step length and cadence. Although several factors have been reported to affect hip moment impulse in the frontal plane during the stance phase (*Inai et al., 2018*; *Inai et al., 2019a*; *Inai et al., 2019b*; *Tateuchi et al., 2020*), the effects of step length and cadence on hip moment impulse in the frontal plane are not known to date. Our results help to understand the gait pattern with low hip moment impulse in the frontal plane.

An increase in stride length leads to an increase in the first peak external hip adduction moment during the stance phase (*Ardestani et al., 2016*). Therefore, we suggest that a decrease in step length may result in a low amplitude of the external hip adduction moment. Furthermore, the hip moment impulse in the frontal plane also decreases because the hip moment impulse in the frontal plane is calculated by the integration of the external hip adduction moment.

*Tateuchi et al. (2020)* examined the effect of gait kinematics of the hip, pelvis, and trunk on external hip adduction moment through hierarchical multiple regression analyses. They reported that the hip moment impulse in the frontal plane deceased with a decrease in stance time, and body mass and stance time accounted for 61% of the variance in hip moment impulse in the frontal plane (*Tateuchi et al., 2020*). Therefore, the effect of stance time on hip moment impulse in the frontal plane is substantial. Additionally, cadence and stance time were significantly negatively correlated in the present study (Table 3). Although the calculation process clarifies that the moment impulse is associated with cadence, that

is, stance time because moment impulse is a time-integrated value, our study is the first to examine the effect of cadence on the hip moment impulse in the frontal plane.

In our study (Table 1), the standardized partial regression coefficients of step length and cadence are 0.63 ($t = 5.24$) and $-0.60$ ($t = -4.58$), respectively. The absolute values of the standardized partial regression coefficients of step length and cadence were the same; therefore, the effects of step length and cadence on hip moment impulse in the frontal plane were similar in the present study. However, the mean walking speed was 1.44 (SD = 0.11) m/s in the present study (mean: 0.82 m/(sHT) ×1.76 m, SD = 0.06 m/(sHT) ×1.76 m, where HT is height). According to *Inai et al. (2019a)*, the effect of a walking speed of approximately 1.3–1.5 m/s on hip moment impulse in the frontal plane was small. In contrast, the effect of walking speed <1.2 m/s on hip moment impulse in the frontal plane is large, and the hip moment impulse in the frontal plane increased exponentially with a decrease in walking speed (*Inai et al., 2019a*). Therefore, the values of the standardized partial regression coefficients of step length and cadence may change in patients with varying slow walking speeds (e.g., absolute values of the standardized partial regression coefficients of step length and cadence are small and large, respectively).

In the present study, the mean hip moment impulse in the frontal plane (not normalized to step length) during walking is 0.50 (SD = 0.17) Nm s/kg. *Tateuchi et al. (2017)* determined that the mean hip moment impulse in the frontal plane during walking at a self-selected speed was 0.41 (SD = 0.13) Nm/kg (mean: 22.7 Nm s/55.2 kg, SD = 7.4 Nm s/55.2 kg). Therefore, the value of the hip moment impulse in the frontal plane in the present study is similar to that reported by *Tateuchi et al. (2017)*. Furthermore, the mean first peak external hip adduction moment during the stance phase in the present study was 1.17 (SD = 0.32) Nm/kg (Table 3), which is similar to that reported by Tateuchi et al. [1.05 (SD = 0.29) Nm/kg (mean: 57.9 Nm/55.2 kg, SD = 16.0 Nm/55.2 kg)] (*Tateuchi et al., 2017*). Therefore, we believe that the values obtained in the present study are quantitatively reasonable.

This study has a few limitations. First, we used a public dataset, and the participants in the dataset were young (*Horst et al., 2019a*; *Horst et al., 2019b*). *Chehab, Andriacchi & Favre (2017)* reported that age does not affect the external hip adduction moment. Therefore, the effect of age on hip moment impulse in the frontal plane may be small. Second, the participants in the present study did not have hip osteoarthritis and were healthy. *Diamond et al. (2018)* reported that the hip joint moments of patients with hip osteoarthritis were different from those of normal participants during walking. Although the results of this study need to be validated in patients with hip osteoarthritis, our findings may be useful in understanding the gait pattern with low hip moment impulse in the frontal plane. The effects of step length and cadence on hip moment impulse in the frontal plane should be further investigated in patients with hip osteoarthritis.

## CONCLUSIONS

Our study revealed that a decrease in step length decreased the hip moment impulse in the frontal plane and a decrease in cadence increased the hip moment impulse in the frontal

plane. These findings may be useful for understanding gait patterns when the hip moment impulse in the frontal plane is low (or high).

## ACKNOWLEDGEMENTS

We would like to thank Editage for English language editing.

### Funding
The authors received no funding for this work.

### Competing Interests
The authors declare there are no competing interests.

### Author Contributions
- Takuma Inai, Tomoya Takabayashi, Mutsuaki Edama and Masayoshi Kubo analyzed the data, prepared figures and/or tables, authored or reviewed drafts of the paper, and approved the final draft.

### Human Ethics
The following information was supplied relating to ethical approvals (i.e., approving body and any reference numbers):

The approval from the ethical committee of the medical association Rhineland-Palatinate in Mainz (Germany) was received (*Horst et al., 2019a*; *Horst et al., 2019b*).

### Data Availability
The data is available at Mendeley Data: Horst, Fabian; Lapuschkin, Sebastian; Samek, Wojciech; Müller, Klaus-Robert; Schöllhorn, Wolfgang I. (2019), ''A public dataset of overground walking kinetics and full-body kinematics in healthy adult individuals'', Mendeley Data, V3, doi: https://doi.org/10.17632/svx74xcrjr.3.

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
