# Peer review of "Effects of step length and cadence on hip moment impulse in the frontal plane during the stance phase"

_PeerJ, doi:10.7717/peerj.11870_

## Round 0.1 · original submission · Major Revisions

While I see some promise in this manuscript, the three reviewers (particularly the first and last reviewer) express some major reservations about whether this paper has the potential to be published.

I strongly encourage you to take on board their suggestions, especially about expanding the breadth of your data analysis as they both rightly point out that the results you presented so far are quite intuitive and may not necessarily have any direct application to osteoarthritis at this point. However, their feedback if you take it on board, definitely has the potential to make this an important paper that will further our understanding in this area.

Reviewer 1 ·

Basic reporting

I agree with the importance of research to prevent progression of hip OA. This paper is generally well organized and the figure is visually valid. However, the association between the risk factor in the progression of hip OA (hip adduction moment impulse) and this study is very weak. The author stated that the results of this study contribute to slowing the progression of hip OA, but this study was a rudimentary analysis of gait, and only presented imaginable results. Daily cumulative hip moment, risk factor for hip OA progression, is product of moment impulse and steps per day. So, increasing cadence alone does not reduce risk.

Experimental design

First, why did this study use healthy and young subjects? There is no reason not to experiment with patients with hip OA. Why the hip moment impulse was normalized with body mass and step length? Step length was also input as an independent variable. This analysis is not appropriate and may confuse the reader. Why did you choose only the hip adduction moment impulse in the frontal plane as a dependent variable? It is understandable to refer to the study of Tateuchi et al. 2017, it is inconclusive that hip adduction moment impulse is the only risk factor for hip OA. It is necessary to justify the sample size why this study required data for as many as 57 people.

Validity of the findings

It is obvious from the calculation process that the moment impulse is related to cadence, that is, stance time because moment impulse is a time-integrated value. It is also widely known that hip adduction moment has bimodal peaks. Unfortunately, I don’t feel any novelty, it’s just a very natural result. Increasing the step length was associated with decrease in the hip moment impulse, but why is that not emphasized?

Additional comments

Research subjects and analysis methods should be reconsidered. And above all, the results of this study do not appear to contribute to slowing the progression of hip OA.

Reviewer 2 ·

Basic reporting

Introduction is relevant to the context.

Structure is clear.

English is clear and the paper is mostly very fluent.
Some detailed comments about language:
L26-28
Please write out this sentence without using the parentheses.
L35
I believe you don’t mean your data as this was public data but probably your findings?
L129-133
Again here I would recommend using full sentences instead of the parentheses to make it more reader friendly. Yes, the clarification is important but it can be written out.
L220
See my previous comment about data vs. findings.

Experimental design

This study is well justified and within the scope of the journal.

Research question is clear, relevant and meaningful.

Specific comments about methods:
Data processing:
Why use 15 of the 20 trials and not all?

Validity of the findings

The findings are well discussed and relevant literature is referenced.

In the conclusions, I would like to see more of an practical application of these findings. So if someone was to do an intervention with a population with hip OA, what kind of adjustments to walking speed and cadence should the clinician advice the patients to make?

Reviewer 3 ·

Basic reporting

The manuscript is well structured and in accordance with a scientific article.

Experimental design

The manuscript used a public dataset that seems to be technically correct.
The statistical methods for the analysis also seems to be correct.
However, in my opinion the authors could explore more the data in to order to properly address their hypotheses.
The issue is that they are investigating the effects of step length and cadence on frontal hip moment without controlling for speed. I will point a possible problem with this approach on the next item.

Validity of the findings

The gait data analyzed are from subjects walking at self-selected speeds with no control of the step length and cadence. So, the gait speed covers a range of values. The authors regressed frontal hip moment with step length and cadence and they found that frontal hip moment is directly correlated with step length and inversely correlated with cadence.
The authors then concluded:
1. "...that a decrease in walking speed does not increase the hip moment impulse...".
Where are the regression results to support this statement?

2. "...a decrease in cadence (not step length) increases the hip moment impulse in the frontal plane."
I think it would be clearer if the authors simply stated the finding that an increase in step length increases the hip moment rather than negating the inverse of the finding.

Additional comments

I think the choice of step length as the focus to discuss the implications to frontal hip moment is somewhat arbitrary.
For example, the authors introduce the notion of cumulative hip moment (product of hip moment impulse in the frontal plane and mean number of steps per day).
So, the finding that an increase in step length leads to an increase in frontal hip moment may not have any implication for osteoarthritis, simply because a larger step length would imply a fewer number of step lengths for a same distance covered on a given day.
I would suggest to the authors to extend their analysis to include the following points:
1. Is frontal hip moment related to gait speed?
2. How much the cumulative hip moment for a hypothetical distance is affected by the increase in step length and speed?

---

## Round 0.2 · Major Revisions

While two reviewers appear satisfied with the revised manuscript, you have not attended adequately to Reviewer's 1 comments. Please look to better address these comments if you wish this paper to be more strongly considered for publication in PeerJ.

Reviewer 1 ·

Basic reporting

I checked the resubmitted manuscript. Unfortunately, I couldn’t find enough revisions to my previous concerns. This study requires reconsideration of the data analysis and subjects. The results of this study (e.g., cadence was associated with the moment impulse) were self-evident and don’t update our knowledge of gait biomechanics. The publication of current manuscript can confuse and mislead readers. The reason is shown below.

Experimental design

Hip moment impulse was normalized by body weight and step length. Nevertheless, the step length was also input as an independent variable in multiple regression analysis for the hip moment impulse. This is clearly an error in the regression analysis. This was pointed out last time, but it hasn't improved.
The authors should include patients with hip OA if they hope that this study will have implications for the prevention of hip OA progression. Research using only the public data of healthy individuals without controlling gait speed will not be able to meet the wishes of the authors. I pointed out it last time; however, no improvement was found.
My previous comments included the following. However, it was not included in the response letter and was ignored.
“It is necessary to justify the sample size why this study required data for as many as 57 people.”

Validity of the findings

It is obvious from the calculation process that the moment impulse is associated with cadence, that is, stance time because moment impulse is a time-integrated value. The current result is naturally expected from the calculation process of the moment impulse. It is essential that the above revisions be made first.

Additional comments

Research subjects and analysis methods should be reconsidered. And above all, the results of this study do not appear to contribute to slowing the progression of hip OA. It’s unfortunate that the previous point wasn’t revised and some comment weren’t taken into account.

Reviewer 2 ·

Basic reporting

No comment.

Experimental design

No comment.

Validity of the findings

No comment.

Additional comments

My previous comments have been addressed adequately. I have no further comments.

Reviewer 3 ·

Basic reporting

The authors have made substantial changes in the manuscript according to my comments. I endorse the publication.

Experimental design

Adequate.

Validity of the findings

Adequate.

---

## Round 0.3 · Minor Revisions

While the second third reviewers were happy with the previously revised manuscript, there are still some points that you have not addressed from reviewer one that have not been completed. Please look to take on board these comments prior to your final submission if you wish this manuscript to be accepted.

---

## Round 0.4 · Minor Revisions

I believe the authors have addressed all of the remaining concerns from the first reviewer. I just have one query prior to this being accepted for publication. Can you please confirm that all the data presented in the results are now based on the randomly selected sample of 31 participants, rather than the full dataset of 57 participants?

---

## Round 0.5 · accepted · Accept

Thank you for the minor clarification and congratulations on your manuscript being accepted for publication in PeerJ.